# A *Salmonella* type III effector, PipA, works in a different manner than the PipA family effectors GogA and GtgA

**Momo Takemura, Takeshi Haneda**👤*, **Hikari Idei, Tsuyoshi Miki, Nobuhiko Okada**

Laboratory of Microbiology, School of Pharmacy, Kitasato University, Minato, Tokyo, Japan

* hanedat@pharm.kitasato-u.ac.jp

**Data Availability Statement:** All relevant data are within the paper and its Supporting Information files.

## Abstract

Nuclear factor-kappa B (NF-κB) plays a critical role in the host defense against microbial pathogens. Many pathogens modulate NF-κB signaling to establish infection in their host. *Salmonella enterica* serovar Typhimurium (*S.* Typhimurium) possesses two type III secretion systems (T3SS-1 and T3SS-2) and directly injects many effector proteins into host cells. It has been reported that some effectors block NF-κB signaling, but the molecular mechanism of the inactivation of NF-κB signaling in *S.* Typhimurium is poorly understood. Here, we identified seven type III effectors—GogA, GtgA, PipA, SseK1, SseK2, SseK3, and SteE—that inhibited NF-κB activation in HeLa cells stimulated with TNF-α. We also determined that only GogA and GtgA are involved in regulation of the activation of NF-κB in HeLa cells infected with *S.* Typhimurium. GogA, GtgA, and PipA are highly homologous to one another and have the consensus zinc metalloprotease HEXXH motif. Our experiments demonstrated that GogA, GtgA, and PipA each directly cleaved NF-κB p65, whereas GogA and GtgA, but not PipA, inhibited the NF-κB activation in HeLa cells infected with *S.* Typhimurium. Further, expressions of the *gogA* or *gtgA* gene were induced under the SPI-1-and SPI-2-inducing conditions, but expression of the *pipA* gene was induced only under the SPI-2-inducing condition. We also showed that PipA was secreted into RAW264.7 cells through T3SS-2. Finally, we indicated that PipA elicits bacterial dissemination in the systemic stage of infection of *S.* Typhimurium via a T3SS-1-independent mechanism. Collectively, our results suggest that PipA, GogA and GtgA contribute to *S.* Typhimurium pathogenesis in different ways.

## Introduction

Nuclear factor-kappa B (NF-κB) is a transcriptional factor that controls cellular processes such as proliferation, differentiation and death, as well as immune responses [1]. In the classical NF-κB signaling pathway, stimulated pattern recognition receptors (PRRs) or tumor necrosis factor (TNF) superfamily receptors lead to the initiation of this signaling pathway, which results in the induction of the degradation of IκBα (nuclear factor of kappa light polypeptide gene enhancer in B-cells inhibitor, alpha) through its phosphorylation or ubiquitination by the

**Funding:** This study was supported in part by Japan Society for the Promotion of Science KAKENHI grants 15K08470 and 19K07543 (to T. H.) and 25293106 and 18K07119 (to N.O.) and by a Kitasato University Research Grant for Young Researchers (2012 to T.H.).

**Competing interests:** The authors have declared that no competing interests exist.

IκB kinase complex [2]. After that, NF-κB forms a hetero-dimer composed of RelA (p65) and NF-κB1 (p50) in the cytoplasm and is finally transported into the nucleus and binds to the promoter region of target genes [2]. The NF-κB signaling pathway plays a central role in the host defense against infection by microbial pathogens, and thus many pathogens modulate NF-κB signaling to establish infection [3].

The *Salmonella enterica* serovar Typhimurium (*S*. Typhimurium) causes inflammatory diarrhea and bacteremia, which are acquired by the oral ingestion of contaminated food. *S*. Typhimurium has two independent type III secretion systems (T3SSs), T3SS-1 and T3SS-2, and effector proteins translocated through these T3SSs are important for their virulence. It has been reported that some effectors secreted through T3SS-1 and/or T3SS-2 modulate the NF-κB signaling cascade. The T3SS-1 effectors SopE, SopE2, and SopB activate the Rho GTPases Rac-1 and Cdc42, which leads to stimulation of the NF-κB pathway. It has been reported that SopD acts cooperatively with SopB [4]. Another T3SS-1 effector, SipA, activates the NF-κB pathway NOD-1/NOD-2 signaling pathway [5]. The translocation of these effectors into the host cells contributes to the invasion of epithelial cells and intestinal inflammation [6,7].

It is also known that *S*. Typhimurium has some effectors which suppress NF-κB activation. A T3SS-2 effector, SseL, has deubiquitinase activity and inhibits by impairing IκBα degradation in *S*. Typhimurium-infected macrophages [8]. AvrA, which is translocated by both T3SS-1 and T3SS-2, also deubiquitinates IκBα, and thereby inhibits NF-κB activation [9]. Two other T3SS-1 and T3SS-2 effectors, GogB and SspH1, have a leucine-rich domain and downregulate the NF-κB pathway by interacting with the host serine/threonine kinase PKN1, and with FBXO22 and Skp1, respectively [10–12].

Recent studies have shown that GtgA, GogA, and PipA function as zinc metalloproteases and target the NF-κB p65 [13,14]. It has also been reported that SseK1, SseK2, and SseK3 are highly related effectors, all of which are N-acetylglucosamine transferases, and block NF-κB signaling by modification of the FAS -associated protein with death domain (FADD) or the TNFR1-associated death domain (TRADD) [15–17]. SpvD, a substrate of T3SS-2, has been shown to interfere with NF-κB activation by blocking of the nuclear transport of NF-κB p65 in macrophages [18]. However, little is known about the mechanism by which NF-κB signaling is downregulated in *Salmonella* pathogenesis.

We conducted the present study to identify effector proteins involved in the blocking of the NF-κB signaling and to investigate the role of these effectors in *Salmonella* pathogenesis. Our findings demonstrated that seven *Salmonella* type III effectors dampen the host immune response by inhibiting NF-κB activation. We observed that NF-κB activation is abrogated by GogA and GtgA, but not by PipA in HeLa cells infected with *S*. Typhimurium. Our study revealed that the PipA family effectors PipA, GogA, and GtgA can dampen NF-κB signaling by cleaving p65, but the role of PipA in the pathogenesis of *S*. Typhimurium differs from the roles of GtgA and GogA.

## Materials and methods

### Animal experiments

The protocols used in this study were approved by the Institutional Animal Care and Use Committee of Kitasato University (protocols J96-1, 17–52, and 20–34). We obtained 8- to 10-week-old female CBA mice from Japan SLC. Mice were inoculated intragastrically with either 0.1 ml of sterile Luria-Bertani (LB) broth or a bacterial culture containing $1\times10^9$ colony-forming units (CFU) of *S*. Typhimurium. Groups of 3–9 mice were euthanized 4 days or 7 days after infection. The cecum was collected and divided into the tip (for histopathological analysis) and the middle (for RNA extraction). Samples from colon contents, mesenteric

lymph nodes (MLNs), or spleen were homogenized in phosphate-buffered saline (PBS), and 10-fold serial dilutions were plated on LB agar plates containing the appropriate antibiotic for determination of the bacterial count.

For histopathology, cecum samples were fixed with 10% buffered-formalin (Mildform 10N; Wako, Osaka, Japan), embedded in paraffin according to standard procedures, sectioned at 5 μm, and stained with hematoxylin and eosin as described previously [19]. The pathologic changes were scored by a slight modification of the blinded examination described by Barthel et al. [20]: namely, the submucosal edema, polymorphonuclear leukocyte (PMN) infiltration, goblet cell numbers, and epithelial damage were examined for a total score of 0–13. More than 3 scores are considered as a sign of inflammation. The scores for PMN infiltration were modified as follows: 0 = <50 PMN/high power field; 1 = 50–70 PMN/high power field; 2 = 71–110 PMN/high power field; 3 = 111–150 PMN/high power field; 4 = >150 PMN/high power field.

RNA was extracted from the mouse cecum with the use of an RNeasy Mini Kit (Qiagen, Hilden, Germany) or an RNA Basic Kit (Nippon Genetics, Tokyo). cDNA was synthesized using TaqMan Reverse Transcription Reagent (Thermo Fisher, San Jose, CA). Real-time polymerase chain reaction (PCR) was performed with the primer pairs described previously [19], using a KAPA SYBR FAST qPCR Master Mix Kit (Kapa Biosystems, Woburn, MA) and analyzed using the comparative Ct method (Applied Biosystems, Foster City, CA). The levels of mRNA expression of target genes were normalized by the levels of *Gapdh* mRNA.

The competitive index (CI) assay has been described previously [21]. For intraperitoneal infection, mice were inoculated with $1 \times 10^4$ CFU of *S*. Typhimurium. For intragastric infection, mice were inoculated with $5 \times 10^7$ CFU of bacteria.

## Bacterial strains and culture conditions

The bacterial strains used are listed in S1 Table. The *S*. Typhimurium wild-type strains used were ATCC 14028 and its spontaneous nalidixic acid-resistant derivative strain SH100 [22]. *E. coli* strain DH5α was used as the host for the construction of plasmids. Unless otherwise indicated, bacteria were grown in LB broth or on LB agar. Antibiotics were added to the media at the following concentrations: ampicillin (100 μg/ml), chloramphenicol (25 μg/ml), nalidixic acid (50 μg/ml), or kanamycin (25 μg/ml). Overexpression of the gene from a *tac* promoter was induced with Isopropyl β-D-1-thiogalactopyranoside (IPTG; 1 μM).

## Construction of mutant strains and plasmids

The bacterial plasmids and primers used are listed in S2 and S3 Tables, respectively. For the construction of *S*. Typhimurium TH1624 (T1), an *invA*::pEP185.2 [23] was transduced from IR715 into SH100 using a P22 phage. Gene deletions of *S*. Typhimurium were constructed by the PCR-based λ Red recombination system [24]. For construction of the double-gene deletion in *S*. Typhimurium, TH1821 (Δ*gogA*Δ*pipA*::Km), a Δ*pipA*::Km mutation from strain TH1772, was introduced by P22 phage, after elimination of the chloramphenicol (Cm) cassette from strain TH1671 (Δ*gogA*::Cm) using pCP20. Using similar methods, other double-deletion strains (TH1681 [Δ*gogA*Δ*gtgA*] and TH1820 [Δ*gtgA*Δ*pipA*::Km]) and multi-deletion mutants (TH1779 [Δ*gogA*Δ*gtgA*Δ*pipA*::Km], TH1722 [Δ*gogA*Δ*gtgA*Δ*pipA*Δ*sseK1*Δ*sseK2*Δ*sseK3*:: KmΔ*steE*::Cm], TH1586 [Δ*sseK1*Δ*sseK2*Δ*sseK3*::Km], TH1766 [*invA*::pEP185.2 Δ*gogA*Δ*gtgA*], and TH1770 [*ssaV*::Cm Δ*gogA*Δ*gtgA*]) were constructed. All mutations were verified by PCR.

*gogA*, *gtgA*, and *pipA* were amplified by PCR and cloned into pTAKN-2 (BioDynamics Laboratory, Tokyo). The PCR product for *gtgA* was digested with BamHI to distinguish between *gogA* and *gtgA*. After TA cloning, these genes were subcloned into a plasmid pFLAG-CTC (Sigma, St. Louis, MO) for the expression of FLAG-fusion proteins in *E. coli* and *S*. Typhimurium, a

plasmid pMW118 (Nippon Gene, Tokyo, Japan) for a complementation assay or a plasmid pEGFP-C1 (Clontech, Mountain View, CA) for expression of EGFP-fusion proteins in mammalian cell lines. Point mutations (on GogA, GtgA, or PipA) were constructed with overlapping primers, TH816 and TH817, TH765 and TH766, or TH846 and TH847, respectively, using a QuikChange Lightning Site-Directed Mutagenesis Kit (Agilent Technologies, Santa Clara, CA). These mutations were verified by DNA sequencing.

pEGFP-NleC [25] and pCMV-FLAG-p65 were provided by Drs. Toru Tobe (Osaka University) and Hiroshi Ashida (Tokyo Medical and Dental University), respectively.

## Cell lines

HeLa cells (ATCC CRL-3126), HEK293T cells (ATCC CCX-2), and RAW264.7 cells (ATCC TIB-71) were routinely cultured in Dulbecco's modified Eagle medium (DMEM; Sigma) supplemented with 10% foetal calf serum (FCS). Cells were grown at 37°C in 5% $CO_2$.

## Transfection

EGFP plasmids or pCMV-FLAG-p65 were transfected into HeLa or HEK293T cells with the reagent PEI-MAX (Polysciences, Warrington, PA) and plasmid in a ratio of 3:2, according to the manufacturer's protocol.

## NF-κB reporter assay

The NF-κB reporter assay was described previously [19]. The NF-κB reporter plasmid pGL4.32 containing an NF-κB response element that drives the transcription of the luciferase reporter gene *luc2P*, the luciferase control plasmid pGL4.74 (Promega, Madison, WI), and the pEGFP-effector plasmid were co-transfected into HeLa cells using Fugene HD (Promega), according to the manufacturer's protocol. Forty-eight hrs later, the cells were stimulated with 10 ng/ml TNF-α (Peprotech, Rocky Hill, NJ) for 30 min. NF-κB activity was calculated using a Dual-Glo Assay System (Promega) according to the manufacturer's instructions.

For the NF-κB reporter assay in HeLa cells infected with *S*. Typhimurium, pGL4.32 and pGL4.74 were transfected into HeLa cells which were seeded at $2.5 \times 10^4$ cells/well in 24-well plates for 24 hrs. Forty-eight hrs after the transfection, the cells were infected with *S*. Typhimurium, which was incubated in LB containing 0.3M NaCl to induce T3SS-1 [26], and the gentamycin killing assay was performed as described previously [19]. After 4 or 20 hrs of infection, the cells were washed three times with PBS and lysed with 1 × Passive Lysis Buffer (Promega). NF-κB activity was calculated as described above.

## Immunofluorescence staining

HeLa cells were seeded at $1.0 \times 10^4$ cells on 13-mm cover glasses in a 24-well plate 24 hrs prior to the transfection. The cells were transfected with pEGFP-effector and stimulated with TNF-α as described above. Cells were fixed with 4% formaldehyde in PBS and permeabilized with 0.1% Triton X-100. The cells were incubated with anti-p65 primary antibody (#8242; Cell Signaling Technology, Beverly, MA) and probed with Alexa Fluor 594 goat anti-rabbit IgG (Molecular Probes, Eugene, OR). Cover glasses were mounted with Vectashield medium with the stain DAPI (Vector Laboratories, Burlingame, CA), and samples were visualized using a fluorescence microscope (VertA1; Zeiss, Jena, Germany).

## Immunoblotting

Immunoblotting was performed according to the standard protocol [22]. Anti-p65 antibody, anti-GFP antibody (04363–24; Nacalai Tesque, Kyoto, Japan), anti-GAPDH (#2118; Cell Signaling Technology), anti-Histone H3 antibody (#4499; Cell Signaling Technology) and anti-GST antibody (04435–84; Nacalai Tesque) were used as primary antibodies, and alkaline phosphatase-conjugated goat anti-mouse and rabbit IgG antibody (Sigma) were used as secondary antibodies. The levels of protein expression were determined by densitometry analysis of immunoblots using Image J software (1.49v; U.S. National Institutes of Health).

## Cell fractionation

HeLa cells were seeded in 6-well plates at $2.0 \times 10^5$ cells/well 24 hrs prior to the transfection or the infection. The transfection of pEGFP-effector into the cells, stimulation of the cells with TNF-α, and infection with *S.* Typhimurium were described above. After the cells were detached with TrypLE Select (Gibco, Grand Island, NY), the cytoplasmic or the nuclear fraction was isolated as described previously [27].

## Gene expression analysis in *S.* Typhimurium by qPCR

Analysis of bacterial gene expression by qPCR has been described previously [21]. S. Typhimurium were grown in LB containing 0.3M NaCl [26] or low phosphate and magnesium-containing medium (LPM, pH 5.8) [28] for induction of T3SS-1 or T3SS-2, respectively. RNA extraction and qPCR were performed as described above. The data were analyzed using the comparative Ct method (Applied Biosystems). Transcription of the target gene was normalized to the levels of *gyrA* mRNA.

## CyaA translocation assay

The CyaA translocation assay was performed as described previously [19]. The secretion of PipA from *S.* Typhimurium into RAW264.7 cells was measured using a cAMP enzyme immunoassay (EIA) kit (Cayman Chemical, Ann Arbor, MI). After 20 hrs of infection, the cells were washed three times with PBS and lysed using the standard protocol recommended by the manufacturer. A cAMP EIA was performed using an iMark microplate reader (BIO-RAD, Hercules, CA).

## LDH assay

Cytotoxicity was assessed by LDH release into the culture medium using a CytoTox 96 Non-Radioactive Cytotoxicity Assay (Promega, Madison, WI). After 20 hrs of infection, the supernatants were collected from infected cells and the $OD_{490}$ was measured using an iMark microplate reader. The relative LDH release was calculated using a CytoTox 96 kit according to the manufacturer's protocol.

## Statistical analyses

Statistical tests were performed with Prism 8 software (GraphPad, La Jolla, CA). One-way ANOVA was used to analyze the relative NF-κB activity in HeLa cells transfected with the pEGFP-effector plasmid. The relative NF-κB activity in HeLa cells infected with *S.* Typhimurium, the gene expression, the bacterial number, and the cytokine expression were analyzed by Student's *t*-test. For the analysis of bacterial numbers, the bacterial numbers (CFU/ml) were transformed logarithmically prior to the statistical analysis. Histopathological scores were

compared by a nonparametric Mann-Whitney *U*-test. The competitive index (CI) was analyzed using a one sample *t*-test. Probability (*p*) values < 0.05 were considered significant.

## Results

### Identification of *Salmonella* type III effectors that interfered with NF-κB activation

For the identification of type III effectors that inactivate the NF-κB pathway, each of the known type III effectors in *S.* Typhimurium was cloned into one copy of the mammalian expression plasmid pEGFP-C1, and the resulting plasmids were transfected into HeLa cells together with both the NF-κB reporter firefly reporter plasmid (pGL4.32) and the control Renilla luciferase plasmid (pGL4.74). After transfection, the HeLa cells were stimulated with TNF-α, and we measured the NF-κB activity by performing a luciferase reporter assay. We confirmed the expression of each GFP-fusion protein in HeLa cells by immunoblotting using anti-GFP antibody or immunofluorescent microscopy (S1 Fig). As shown in earlier reports, transfection with NleB1, which is a type III effector in enteropathogenic or enterohemorrhagic *E. coli* (EPEC or EHEC), led to a decrease in the NF-κB activity compared to transfection of cells with an empty vector [29–31]. In contrast, a *Salmonella* effector, SpvC, was shown in our previous study to be a phosphothreonine lyase that does not affect NF-κB signaling [29–31]. As with the NleB1 transfection, the relative NF-κB activity of HeLa cells transfected with seven plasmids was significantly reduced compared to that in the cells transfected with SpvC (Fig 1). Those seven plasmids contained the *gogA*, *gtgA*, *pipA*, *sseK1*, *sseK2*, *sseK3*, or *steE* gene.

### GogA and GtgA, but not PipA, attenuate the NF-κB response in *Salmonella*-infected HeLa cells

To study the effects of these effectors on the NF-κB pathway during *S.* Typhimurium infection, we constructed an *S.* Typhimurium strain lacking all seven of the identified genes (*gogA*, *gtgA*, *pipA*, *sseK1*, *sseK2*, *sseK3* and *steE*) and infected HeLa cells transfected with both pGL4.32 and

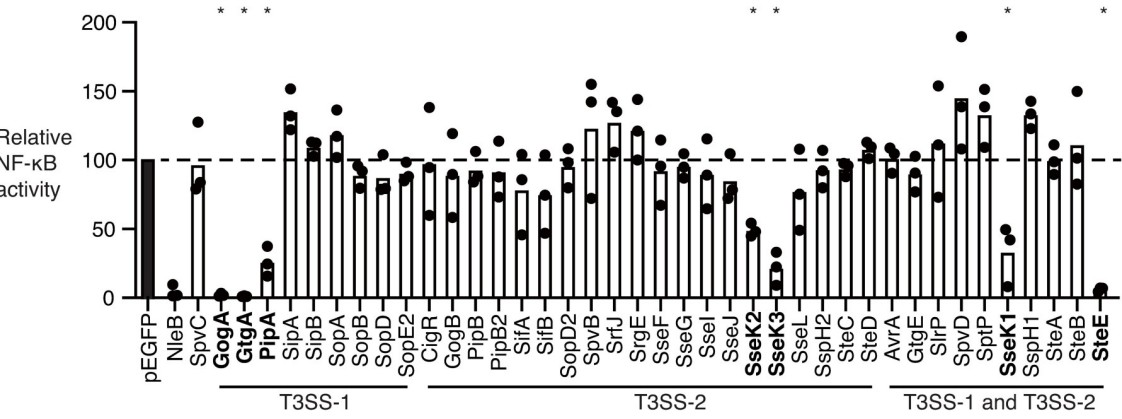

**Fig 1. A comprehensive analysis of the *Salmonella* type III effectors that inhibit NF-κB activation.** HeLa cells were transfected with the indicated pEGFP-effector fusion plasmid together with pGL4.32 and pGL4.74. After 48 hrs, the cells were stimulated with TNF-α (10 ng/ml) and further cultured for 30 min, and the NF-κB activity was measured. The relative NF-κB activity is the value of the NF-κB activity of HeLa cells transfected with the pEGFP-effector fusion plasmid relative to that of the HeLa cells transfected with the empty vector, which was taken as 100. One-way ANOVA analysis (Dunnett's multiple comparison test against the corresponding HeLa cells transfected with pEGFP-SpvC) was performed for statistical analysis. Individual data are shown as a scatter plot, and bars are the mean. Asterisks indicate statistically significant differences (*p* < 0.05). The effectors that inhibit NF-κB activation are shown in bold.

pGL4.74. At 4 or 20 hrs after infection, we measured the luciferase levels and calculated the relative NF-κB activity. This strain exhibited greater NF-κB activity than the wild-type *S.* Typhimurium at 4 hrs after infection, but the same level as the wild-type at 20 hrs after infection (Fig 2A). This indicated that at least one of these genes is related to the inhibition of NF-κB activation in the *S.* Typhimurium-infected cells at 4 hrs after infection.

Next, we examined which of these seven effectors is required for suppression of NF-κB activation. GogA, GtgA and PipA belong to the PipA family of effectors, which interfere with NF-κB signaling by cleaving p65 [13,14]. SseK1, SseK2, and SseK3 are highly related effector proteins, and they share a high degree of homology with NleB1, an EHEC or EPEC T3SS effector that functions as an N-acetylglucosamine (GlcNAc) transferase to modify death domain proteins and inhibit NF-κB activation [32–34]. SteE is an effector through T3SS-1 and T3SS-2 [35]; Recent reports have shown that SteE (also known as SarA) promotes secretion of IL-10 from infected B cells [36] and induces anti-inflammatory macrophage polarization [37]. However, it is not clear whether these phenotypes are related to the inhibition of NF-κB activation.

To determine which genes were related to NF-κB suppression, we further constructed the strains TH1779 (Δ*gogA*Δ*gtgA*Δ*pipA*::Km), TH1586 (Δ*sseK1*Δ*sseK2*Δ*sseK3*::Km) and TH1761 (Δ*steE*::Cm). In contrast to HeLa cells infected with Δ*sseK1*Δ*sseK2*Δ*sseK3*::Km or Δ*steE*::Cm, only Δ*gogA*Δ*gtgA*Δ*pipA*::Km exhibited greater NF-κB activity than the wild-type *S.* Typhimurium (Fig 2B). These results show that GogA, GtgA, and/or PipA inhibited NF-κB signaling in the *S.* Typhimurium-infected HeLa cells.

GtgA, GogA, and PipA proteins have high homology to one another. These proteins belong to the peptidase M85 Pfam family (PF13678). Members of this family have a consensus zinc metalloprotease HExxH motif. GtgA and GogA possess the HEVVH amino acid (aa) sequences between aa 182 and 186, and PipA has the motif in aa residues 180–184. Several groups have reported that NleC, a type III effector from EPEC or EHEC, contains an HExxH motif and inhibits NF-κB signaling by a proteolytic cleavage of p65 [25,38–42]. It has also been shown that GtgA, GogA, and PipA collectively function as zinc metalloproteases and target the NF-κB p65 [13,14]. However, the roles of individual effectors in *S.* Typhimurium pathogenesis are not yet clear.

To investigate which effector facilitates the loss of NF-κB activation in the Δ*gogA*Δ*gtgA*Δ*pipA*::Km mutant strain, we analyzed three single mutant strains, TH1671 (Δ*gogA*::Cm), TH1678 (Δ*gtgA*::Cm), and TH1772 (Δ*pipA*::Km), three double-deleted mutant strains, TH1681 (Δ*gogA*Δ*gtgA*), TH1821 (Δ*gogA*Δ*pipA*::Km), and TH1820 (Δ*gtgA*Δ*pipA*::Km), and a triple-deletion mutant strain, TH1779 (Δ*gogA*Δ*gtgA*Δ*pipA*::Km). The NF-κB activity in HeLa cells infected with Δ*gogA*::Cm, Δ*gogA*Δ*gtgA*, or Δ*gogA*Δ*gtgA*Δ*pipA*::Km was significantly higher than that of HeLa cells infected with the wild-type *S.* Typhimurium (Fig 2C). Surprisingly, the increased level of NF-κB activity in HeLa cells infected with Δ*gogA*Δ*gtgA* was the same as that of the cells infected with Δ*gogA*Δ*gtgA*Δ*pipA*::Km (Fig 2C).

Next, to reveal the role of these effectors in the HeLa cells infected with *S.* Typhimurium, we constructed Δ*gogA*Δ*gtgA* strains complemented with a plasmid expressing flag-tagged *gogA*, *gtgA*, or *pipA*. The expressions of each gene from the plasmid were driven by the *tac* promoter with its inducer, isopropyl β-D-1-thiogalactopyranoside (IPTG) and were monitored by immunoblotting using anti-FLAG antibody (S2 Fig). The introduction of a plasmid expressing *gogA*, *gtgA*, or *pipA* into the Δ*gogA*Δ*gtgA* strain resulted in a decrease in the activation of NF-κB compared to the introduction of an empty plasmid, pMW118 (Fig 2D). However, the phenotype of Δ*gogA*Δ*gtgA* could not be restored by induction of the plasmid pGogA_{H182Y}, pGtgA_{H182Y}, or pPipA_{H180Y}; these plasmids have a histidine-to-tyrosine point mutation at the first histidine in the consensus zinc metalloprotease motif on GogA, GtgA, or PipA,

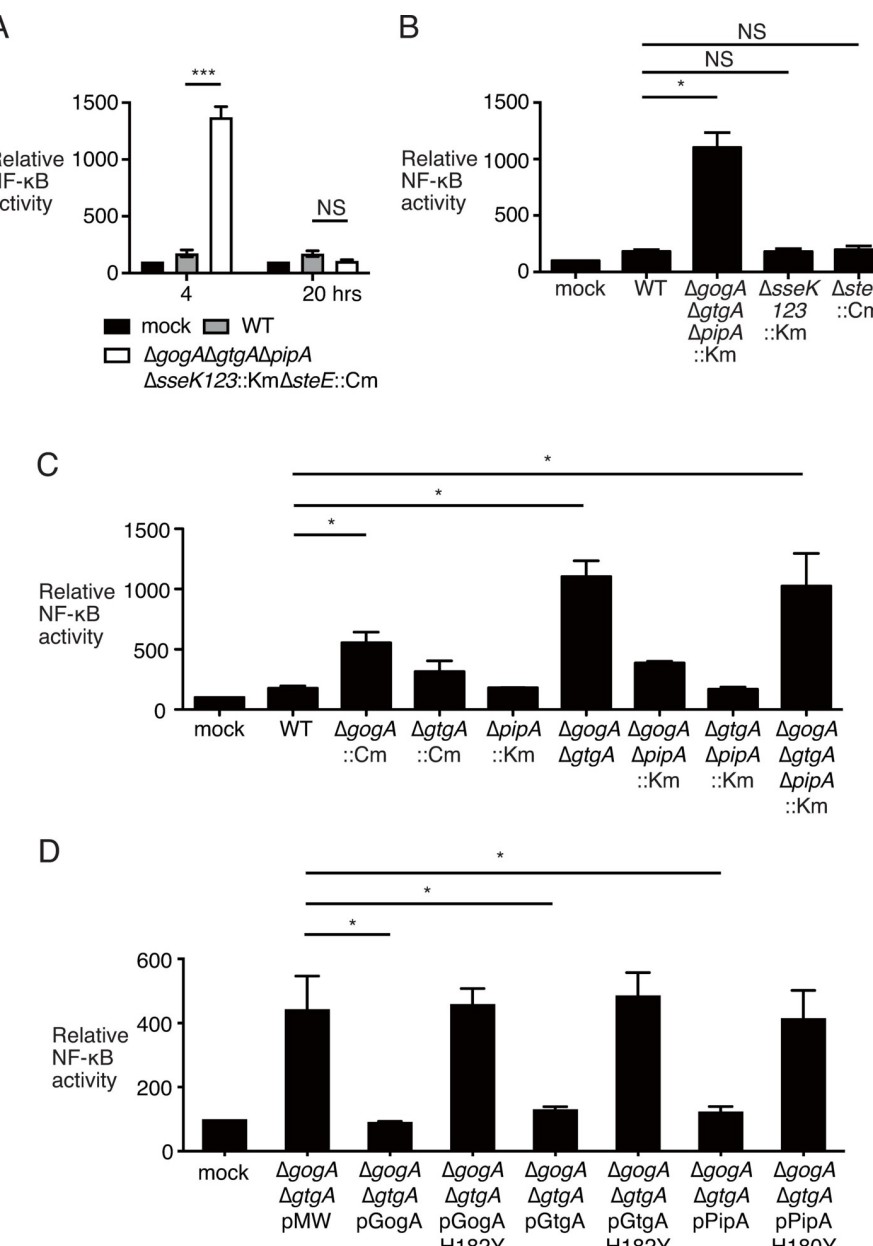

**Fig 2. GogA and GtgA, but not PipA, dampen NF-κB activation in *S*. Typhimurium-infected HeLa cells.** HeLa cells were transfected with pGL4.32 and pGL4.74. After 48 hrs, the cells were infected with T3SS-1-induced *S*. Typhimurium strains at a multiplicity of infection of 50 and the gentamycin killing assay was performed. After 4 hrs (A–D) and 20 hrs (A) of infection, the cells were washed three times with PBS and lysed with 1 × Passive Lysis Buffer. The relative NF-κB activity is the value of the NF-κB activity of HeLa cells infected with the indicated *S*. Typhimurium strains relative to those of mock-infected cells, which were taken as 100. Data represent the mean and SD of at least three independent experiments. Two-tailed Student's *t*-test was performed for statistical analysis. Asterisks indicate statistically significant differences (***$p < 0.001$, *$p < 0.05$ vs. wild-type infection [A-C] or Δ*gogA*Δ*gtgA* pMW infection [D]). NS, not statistically significant.

respectively (Fig 2D). Collectively, these data suggest that GogA and GtgA, but not PipA, attenuate the NF-κB response in the *S*. Typhimurium-infected HeLa cells, and that PipA can dampen the NF-κB activation only when overexpressed in the Δ*gogA*Δ*gtgA* strain.

## GogA and GtgA cleave p65 in *S.* Typhimurium-infected HeLa cells

We next determined whether NF-κB p65 is cleaved in HeLa cells infected with the *S.* Typhimurium wild-type or mutant strains. We observed cleaved p65 in HeLa cells infected with the wild-type and all three of the single mutants, but not in HeLa cells infected with the Δ*gogA*Δ*gtgA*Δ*pipA*::Km mutant at 2 hrs after infection (Fig 3A). In HeLa cells infected with double mutants, Δ*gogA*Δ*pipA*::Km and Δ*gtgA*Δ*pipA*::Km could cleave p65, but Δ*gogA*Δ*gtgA* could not (Fig 3B). However, cleaved p65 was not detected in HeLa cells infected with the wild-type or any of the mutants at 16 hrs after infection (S3 Fig). These findings indicate that GogA and GtgA act in a redundant manner to digest the p65 protein in the early stage of infection with *S.* Typhimurium. To confirm this finding, we infected HeLa cells with Δ*gogA*Δ*gtgA* carrying a plasmid cloned from a wild-type or a mutant type of *gogA*, *gtgA*, or *pipA*. As expected, the phenotype of Δ*gogA*Δ*gtgA* was complemented by pGogA, pGtgA, or pPipA, but not by pGogA$_{H182Y}$, pGtgA$_{H182Y}$, or PipA$_{H180Y}$ (Fig 3C). These results are consistent with the finding that GogA and GtgA, and even PipA, block NF-κB signaling in HeLa cells infected with *S.* Typhimurium, when at least one of these effectors is expressed in the Δ*gogA*Δ*gtgA* strain.

## PipA family effectors, GogA, GtgA, and PipA, possess the zinc metalloprotease activity

It has been reported that GogA, GtgA, and PipA inhibit NF-κB signaling by the cleavage of p65 [14,43]. However, our present results demonstrated that GogA and GtgA, but not PipA, blocked the NF-κB activation in the *S.* Typhimurium-infected cells. To confirm that GogA,

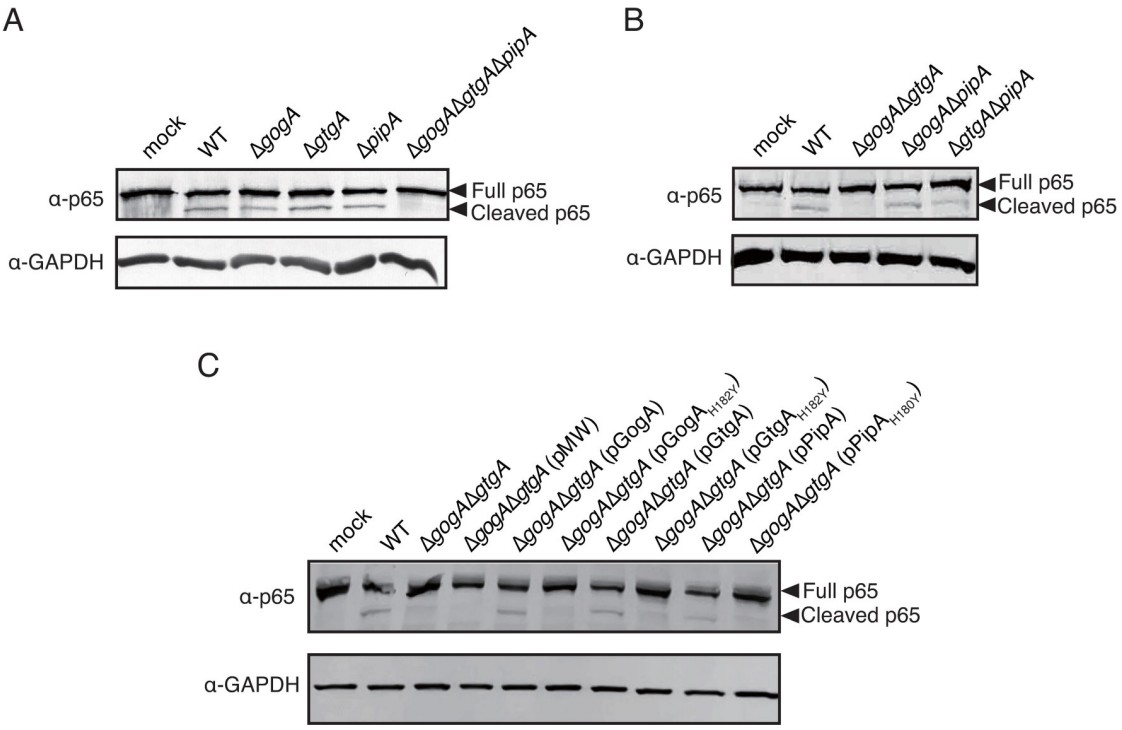

**Fig 3. GogA and GtgA, but not PipA, cleave NF-κB p65 in *S.* Typhimurium-infected HeLa cells.** HeLa cells were infected with the indicated *S.* Typhimurium strains and the gentamycin killing assay was performed. After 2 hrs of infection, the cells were detached with TrypLE Select and the cytoplasmic fraction was isolated. The full-length p65 (Full p65) or cleaved p65 (Cleaved p65) in the cytoplasmic fraction was detected by immunoblotting using an anti-p65 antibody. Immunoblots are representative of at least three independent experiments.

GtgA, and/or PipA act as zinc metalloproteases, we constructed the mammalian expression plasmids pEGFP-GtgA, pEGFP-GogA, and pEGFP-PipA, which have a histidine-to-tyrosine point mutation at the first histidine on GogA, GtgA and PipA, respectively (pEGFP-GtgA$_{H182Y}$, pEGFP-GogA$_{H182Y}$ and pEGFP-PipA$_{H180Y}$), and we transfected the plasmids into HEK293T cells. The p65 signal was reduced in the cells that expressed wild-type GogA, GtgA or PipA, but not in the cells that expressed the point mutants (S4 Fig).

To facilitate the enzymatic activity of these proteins, we determined the amounts of cytoplasmic p65 in HEK293T cells transiently expressing the wild-type or point mutants of GogA, GtgA, or PipA by using an anti-p65 antibody. Full and cleaved p65 were observed in the cells expressing the wild-type GogA, GtgA, or PipA, as well as those expressing NleC, but cleaved p65 was not detected in the cells transfected with the point mutants (S5 Fig). Taken together, these data indicate that GogA, GtgA, and PipA exhibit zinc metalloprotease activity, which is consistent with the findings of previous studies.

## PipA functions as a T3SS-2 effector

As shown in Figs 2 and 3, the phenotype of a Δ*gogA*Δ*gtgA* strain was complemented by overexpression of *gogA*, *gtgA*, or *pipA*. We therefore reasoned that PipA may not be expressed in HeLa cells infected with *S*. Typhimurium. Originally, PipA was identified as a virulence factor which is required for *Salmonella* enteropathogenicity [44]. It has also been reported that PipA is secreted under T3SS-1-inducing conditions [35]. These reports suggest that PipA is a T3SS-1 effector. However, PipA is located in an operon with a gene encoding the T3SS-2 effector PipB (S6 Fig) [44,45]. Moreover, it has been indicated that PipA is up-regulated in macrophages at the late stage of infection [46]. Hence, we hypothesized that PipA functions as a T3SS-2 effector. Previous studies have shown that the expressions of T3SS-1 and its effectors are induced in LB medium containing 0.3M NaCl, while the expressions of T3SS-2 and its effectors are induced in low phosphate and magnesium-containing medium (LPM; pH 5.8) [26,28]. To determine whether the *pipA* gene was expressed under the T3SS-1-inducingand/or T3SS-2-inducing conditions, the mRNA levels of the *gogA*, *gtgA* (these levels were indistinguishable and are given as *gogA*/*gtgA* hereinafter), and *pipA* genes were assessed by qRT-PCR. Under the T3SS-1-inducing conditions, a SPI-1 positive regulator, *invF*, and *gogA*/*gtgA* were induced, but a SPI-2 positive regulator, *ssrB*, and *pipA* were not (Fig 4A). On the other hand, under the T3SS-2-inducing conditions, *ssrB*, *gogA*/*gtgA*, and *pipA* were induced (Fig 4B).

Next, to confirm that PipA is secreted through T3SS-2, we constructed a gene fusion protein between *pipA* and a calmodulin-dependent adenylate cyclase (*cyaA*), and performed a CyaA translocation assay. The PipA–CyaA fusion protein was then produced in the *S*. Typhimurium wild-type or in a mutant strain lacking the *invA* (T3SS-1-deficient mutant, T1) or *invA* and *ssaV* genes (T3SS-1 and T3SS-2-deficient mutant, T1T2). To confirm the function of T3SS-2 in the strains tested, we used SseJ, a component of the T3SS-2 translocon, fused to CyaA (SseJ–CyaA) as a positive control and cultivated each strain under T3SS-2-inducing conditions. To confirm the T3SS-2-dependent translocation of PipA into host cells, cells of a mouse macrophage-like cell line, RAW264.7 (RAW), were infected with the *S*. Typhimurium wild-type or the mutant strain T1 or T1T2. After 20 hrs of infection, a higher level of translocation of PipA–CyaA was induced in the wild-type and T1-deficient strains than in either the T1-deficient or T2-deficient strain (Fig 4C). These data suggest that PipA can be behave as a T3SS-2 effector. To further characterize the function of PipA as a T3SS-2 effector, we analyzed the contribution of PipA to intracellular survival and induction of cell death in macrophage-infected *S*. Typhimurium. The *invA*::pEP185.1

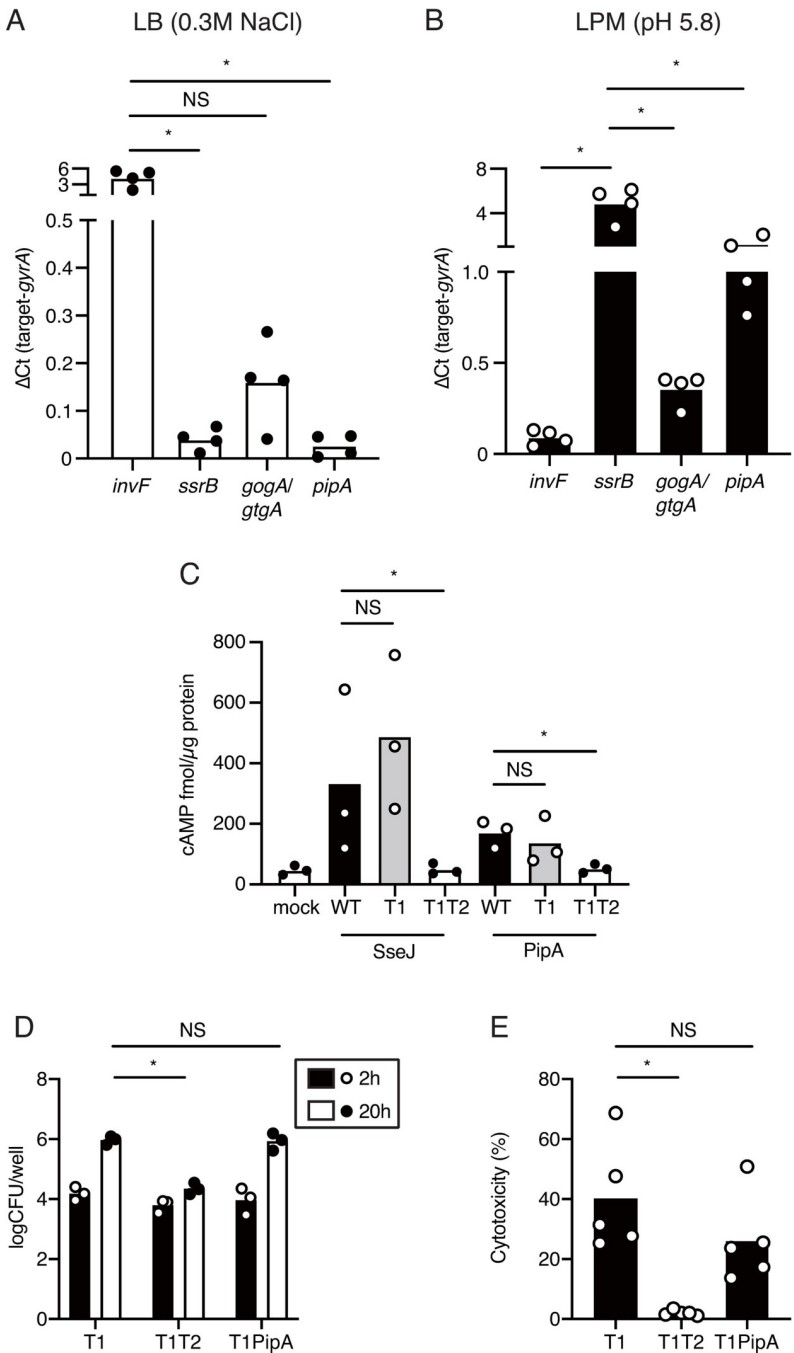

**Fig 4. PipA is a T3SS-2 effector.** (A-B) Relative expression levels of genes in *S*. Typhimurium grown in LB with 0.3M NaCl (SPI-1-inducing condition; A) or low phosphate and magnesium-containing medium (LPM) pH 5.8 (SPI-2-inducing condition; B) were measured by qRT-PCR. (C) Translocation of PipA–CyaA was analyzed at 20 hrs after infection of RAW cells with the indicated *S*. Typhimurium strains. (D) Numbers of bacteria recovered from RAW cells infected with the indicated *S*. Typhimurium mutant strains at 2 and 20 hrs after infection. (E) LDH release from RAW cells infected with the indicated *S*. Typhimurium mutant strains at 20 hrs after infection. A two-tailed Student's *t*-test was performed for statistical analysis. Individual data are shown as a scatter plot, and bars indicate the means. Asterisks indicate statistically significant differences ($P < 0.05$). NS, not significant.

Δ*pipA*::Km strain resulted in the same level of intracellular survival and induction of cell death in RAW cells as the T1 strain (Fig 4D and 4E).

## PipA contributes to systemic dissemination of S. Typhimurium in mice

The *S*. Typhimurium Δ*gogA*Δ*gtgA*Δ*pipA* mutant elicits severe inflammation in the cecum of *Slc11a1* (previously *Nramp1*) $^{+/+}$ mice compared to *Slc11a1*$^{-/-}$ mice at 4 days after infection [14]. We therefore assessed the inflammation in the cecum of *Slc11a1*$^{+/+}$ mice (CBA) infected with the *S*. Typhimurium wild-type, or the Δ*gogA*Δ*gtgA* or Δ*gogA*Δ*gtgA*Δ*pipA*::Km mutant. No difference was observed in the number of bacteria in the cecum, MLNs, or spleen, or in the histopathological score in the cecum between the *S*. Typhimurium wild-type, Δ*gogA*Δ*gtgA*, and Δ*gogA*Δ*gtgA*Δ*pipA*::Km strains at 4 days after infection (S7 Fig). In the cecum of CBA mice infected with the *S*. Typhimurium wild-type strain at 4 days after infection, inflammation was not sufficiently induced (S7 Fig). We thus repeated the experiments at 7 days after infection, and we found that Δ*gogA*Δ*gtgA* tended to have higher colonization in the colon contents than the wild-type (Fig 5A). However, there were no significant differences in the number of CFUs in the colon contents, MLNs, or spleen among mice infected with the *S*. Typhimurium wild-type, Δ*gogA*Δ*gtgA*, and Δ*gogA*Δ*gtgA*Δ*pipA*::Km strains (Fig 5A). Moreover, at 7 days after infection, the Δ*gogA*Δ*gtgA* or Δ*gogA*Δ*gtgA*Δ*pipA*::Km strains tended to induce a greater level of inflammation in the cecum of mice compared to the wild-type strain, but no differences in the histopathological score or inflammatory cytokines in the cecum were detected among mice infected with the three strains (Fig 5B and 5C).

The survival of BALB/c (*Slc11a1*$^{-/-}$) mice orally infected with the *S*. Typhimurium Δ*pipA* mutant was significantly prolonged compared to that of mice infected with the wild-type [45]. As shown in Fig 4, PipA functions as a T3SS-2 effector. Therefore, we hypothesized that the zinc protease activity of PipA secreted from a T3SS-2 was related to systemic infection of *S*. Typhimurium in mice. To examine this hypothesis, we first analyzed the virulence of the T3SS-1-and *pipA*-deficient mutant using a competitive index (CI) assay. C57BL/6 mice were intragastrically inoculated with *invA*::pEP185.1 (T1) and *invA*::pEP185.1 Δ*pipA*::Km (T1 PipA). The average of the CIs of colon contents from these mice was not significantly different from 1 (logCI = 0) (Fig 6A). On the other hand, the average of the CIs of the mesenteric lymph nodes (MLNs) and spleens of the mice were significantly lower than 1 (Fig 6A). However, the average of the CIs of the spleens from mice intraperitoneally infected with these *S*. Typhimurium strains was not significantly different from 1 (Fig 6B). These data show that T1 PipA reduces the systemic dissemination of bacteria in a mouse typhoid model.

Finally, we wanted to know whether the zinc metalloprotease activity of PipA contributes to systemic infection in mice. However, the *S*. Typhimurium Δ*gogA*Δ*gtgA* strain carrying either pMW-PipA or pMW-PipA$_{H180Y}$, shown in Figs 2 and 3, was not well suited for studies of the role of the enzymatic activity of PipA, because plasmid instability was predicted during infection of mice [23]. Therefore, to study the role of PipA *in vivo*, we constructed *S*. Typhimurium strains carrying the wild-type *pipA* gene or *pipA*$_{H180Y}$ gene, which expresses the inactivated form of zinc metalloprotease, inserted in the *phoN* gene on the genome of *S*. Typhimurium (*phoN*::*pipA* or *phoN*::*pipA*$_{H180Y}$). Successful complementation was achieved for T1 PipA by the expression of the wild-type *pipA* gene, but not the *pipA*$_{H180Y}$ mutant (Fig 6C). These results suggest that the zinc protease activity of PipA through T3SS-2 is conducive to dissemination of *S*. Typhimurium to systemic sites of infection.

## Discussion

Here, we report that *Salmonella* effector proteins, i.e., GogA, GtgA, PipA, SseK1, SseK2, SseK3, and SteE, dampen NF-κB signaling. Although all of these effectors except SteE have previously been shown to inhibit NF-κB activation, our present findings demonstrate that only GogA and GtgA are related to the blocking of the NF-κB signaling in HeLa cells

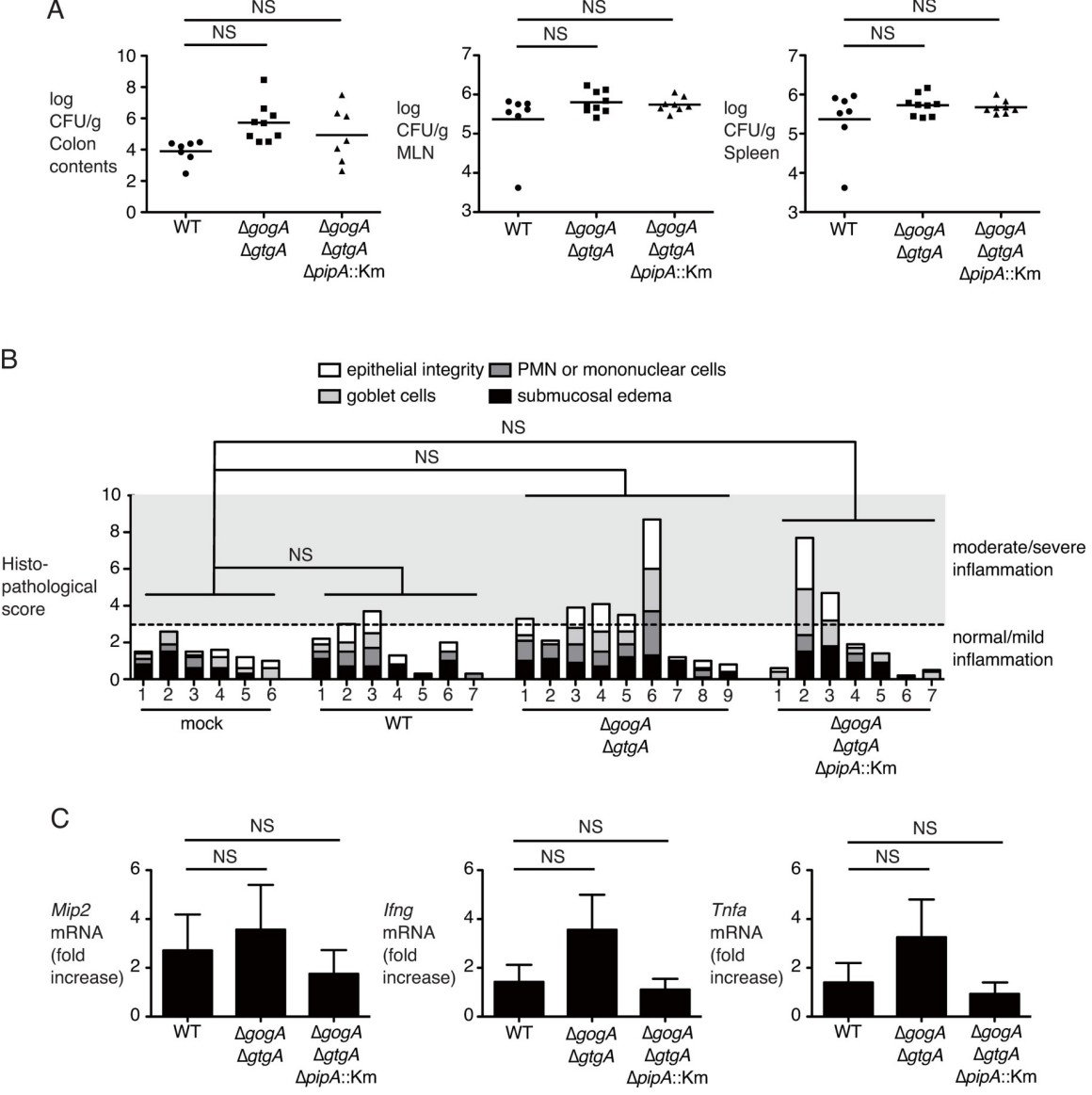

**Fig 5. The Δ*gogA*Δ*gtgA* and the Δ*gogA*Δ*gtgA*Δ*pipA* strains do not increase the virulence of *S*. Typhimuirum in the infected CBA mice.** CBA mice were infected intragastrically with 1×10⁹ CFU of the indicated *S*. Typhimurium strains. (**A–C**) Bacterial numbers recovered from the colon contents, mesenteric lymph nodes (MLNs) and spleen (A), histopathological scores in sections of the cecum (B) and transcript levels of *Mip2*, *Ifng*, and *Tnfa* (C) at 7 days after infection are shown. In (A), individual data are shown as a scatter plot, and bars are the mean. In (B), each bar represents the combined scoring results for a single mouse. In (C), data represent the mean and SD. A two-tailed Student's *t*-test (A and C) or a non-parametric Mann-Whitney's U test (B) was performed for statistical analysis. NS, not significant.

infected with *S*. Typhimurium. In Fig 2C, the NF-κB activity of HeLa cells infected with the Δ*gogA* strain was higher than that of HeLa cells infected with the wild-type strain but lower than that of HeLa cells infected with the Δ*gogA*Δ*gtgA* or Δ*gogA*Δ*gtgA*Δ*pipA* strain. Further, the Δ*gogA* strain, as well as the Δ*gtgA* and Δ*pipA* strains, was capable of cleaving NF-κB p65 (Fig 3A). These findings could indicate that not only the Δ*gtgA* and Δ*pipA* strains, but also the Δ*gogA* strain dampens the NF-κB signaling, and that GogA and GtgA play a redundant role in cleaving p65.

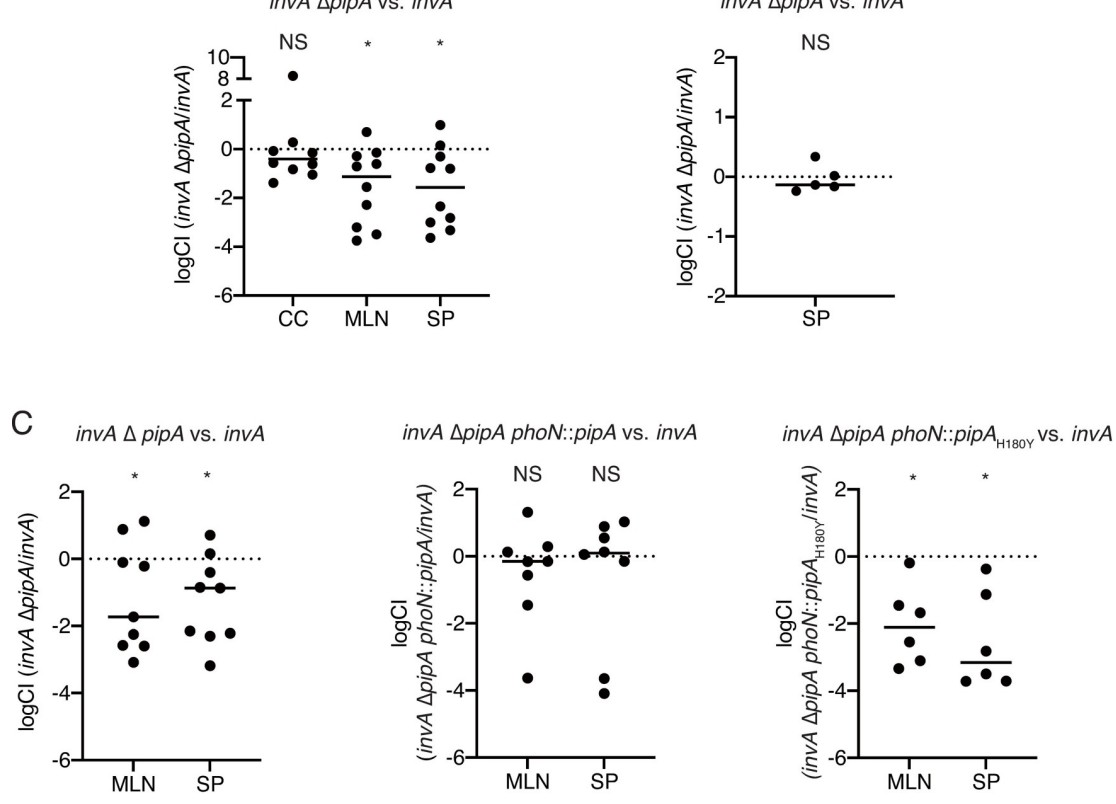

**Fig 6. PipA contributes to bacterial dissemination in systemic infection with S. Typhimurium.** (A-B) Competitive index (CI) of the colon contents (CC), mesenteric lymph nodes (MLNs) and spleens (SP) of mice infected with *S.* Typhimurium *invA*:: pEP186.5 (T1) and *invA*::pEP186.5 Δ*pipA*::Km (T1 PipA) intragastrically (A) and intraperitoneally (B). (C) CI of the MLNs and SP from mice intragastrically infected with *S.* Typhimurium T1 and T1 PipA, *invA*::pEP186.5 Δ*pipA*::Km *phoN*::*pipA*, or *invA*:: pEP186.5 Δ*pipA*::Km *phoN*::*pipA*$_{H180Y}$. One sample *t*-test (hypothetical value: 1 [logCFU = 0]) was performed for statistical analysis. Individual data are shown as a scatter plot, and bars indicate the means. Asterisks indicate statistically significant differences ($P < 0.05$). NS, not significant.

NF-κB p65 was cleaved in HeLa cells infected with the *S.* Typhimurium wild-type or Δ*ssaV*::Cm (T2) strain, but not in those infected with the Δ*gogA*Δ*gtgA* or Δ*invA*::pEP185.2 (T1) strain (S8 Fig). This result indicates that this phenotype observed in HeLa cells at 4 hrs post-infection is dependent on T3SS-1. In contrast, SseK1, SseK2, SseK3, and/or SteE may function at a late stage of infection, since these effectors were translocated through T3SS-2 or both T3SS-1 and T3SS-2 [32,33,35]. Several studies have reported that other effectors of T3SS-2 or of both T3SS-1 and T3SS-2—i.e., AvrA, GogB, SpvD, SseL, SspH1, and SteA—inhibit NF-κB activation [8–12,47]. Interestingly, as shown in Fig 2A, the level of NF-κB activity was low In HeLa cells infected with the *S.* Typhimurium wild-type at 20 hrs after infection, and similar to the levels at 4 hrs after infection. This indicates that the NF-κB signaling is suppressed in HeLa cells during the late stage of infection. However, the level of NF-κB activity In HeLa cells infected with the Δ*gogA*Δ*gtgA*Δ*pipA*Δ*sseK1*Δ*sseK2*Δ*sseK3*Δ*steE* strain after 20 hrs of infection was similar to that in the HeLa cells infected with the wild-type. Moreover, cleaved p65 was not detected in HeLa cells infected with the wild-type after 16 hrs of infection (S3 Fig). elHeThese data suggest that other T3SS-2 effectors inhibit the NF-κB signaling pathway in a way that is different from cleaving p65. Further studies will be needed to understand the role of suppression of NF-κB activation in the late stage of *S.* Typhimurium-infection.

PipA, GogA, and GtgA were reported to cleave NF-κB p65 and inhibit the NF-κB response in epithelial cells and fibroblasts [13,14]. Our present findings showed that PipA does not contribute to the dampening of the NF-κB activation in *Salmonella*-infected HeLa cells. However, the Δ*gogA*Δ*gtgA* (pPipA) strain, which overexpresses PipA from the plasmid, dampens the NF-κB activation. Cleaved p65 was also observed in HeLa cells transfected with PipA. These facts suggest that the PipA protein can cleave p65 but does not appear to do so at the times measured in HeLa cells during S. Typhimurium infection. PipA, GogA and GtgA are classified as members of the PipA family because of the high similarity among their amino acid sequences, but there are some differences between these three proteins. First, the amino acid sequence of PipA is homologous to those of GtgA and GogA, but the similarity of the amino acid sequence between PipA and GtgA or between PipA and GogA is slightly lower than that between GtgA and GogA (S6 Fig). The promoter region of the *pipA* gene and five N-terminal amino acids of the PipA protein are also different from those of the *gtgA* or *gogA* gene and the GtgA or GogA protein, respectively (S6 Fig). These findings may indicate that the patterns of protein expression and secretion for PipA differ from those for GogA and GtgA. Consistent with these findings, we here showed that PipA was not expressed under the SPI-1-inducing conditions.

The *gogA* and *gtgA* genes but not the *pipA* gene are located on Gifsy-1 or Gifsy-2 prophages integrated in the genome of S. Typhimurium. Another T3SS-1 and T3SS-2 effector, GogB, which suppresses NF-κB signaling, is also located on the Gifsy-1 prophage [12]. SteE, which was identified in the present study, is also encoded on this prophage. Interestingly, the cleavage of p65 was not observed in HeLa cells infected with S. Enteritidis, which causes a food-borne disease (as does S. Typhimurium) (S8 Fig). In the genome of the highly virulent strain S. Typhimurium UK-1, which was isolated from an infected horse, one more copy of metalloprotease (STMUK_2657) is located on the Gifsy-1 prophage compared to *gogA* (STMUK_2648), *gtgA* (STMUK_0993), and *pipA* (STMUK_1056) [48]. STMUK_2657 is a homolog of NleC, which is a type III effector in EHEC or EPEC and interferes with NF-κB activation by cleaving p65 [49]. Thus, the inactivation of the NF-κB signaling pathway by these effectors may be important for the virulence of S. Typhimurium.

PipA belongs to *Salmonella* pathogenicity island 5 (SPI-5), which is conserved in all *Salmonella* serovars [44]. In bovine ligated ileal loops infected with S. Dublin, PipA is related to the induction of intestinal secretory and inflammatory responses [44]. PipA also contributed to intestinal colonization and systemic dissemination in 1-day-old chickens [50]. These findings suggest a possible role of PipA for virulence in not only the serovar Typhimurium, but also other serovars. In this study, we indicated that the zinc metalloprotease activity of PipA plays an important role for the systemic infection of S. Typhimurium, but were not be able to identify the target for PipA. In EPEC or EHEC, a type III effector, NleD, was identified as a zinc metalloprotease and was reported to cleave JNK and p38 MAPKs, but not NF-κB [42]. This might indicate that PipA digests another target in *Salmonella* infection. Further studies are needed to identify substrates for the PipA zinc metalloprotease and to determine the role of PipA in *Salmonella* pathogenesis.

Sun et al. showed that deletions of the *gogA*, *gtgA*, and *pipA* genes increase intestinal inflammation in C57/BL6 *Slc11a1* (previously *Nramp1*) $^{+/+}$ mice, but not wild-type C57/BL6 mice (*Slc11a1*$^{-/-}$) [14]. Slc11a1 encodes the divalent metal iron transporter that confers resistance to S. Typhimurium infection [51]. Some groups have shown that S. Typhimurium leads to inflammation in CBA mice (*Slc11a1*$^{+/+}$), approximately 10 days after infection [52,53]. In the present study, therefore, we used CBA mice, but we did not observe any difference in the histopathological score or the production of inflammatory cytokines between the mice infected with the S. Typhimurium wild-type strain and those infected with the Δ*gogA*Δ*gtgA*Δ*pipA*

mutant. We noticed that the number of bacteria in colon contents of CBA mice infected with the wild-type strain SH100 ($10^4$ CFU/g) at 7 days after infection was lower than that in the feces of mice infected with strain IR715 ($10^8$ CFU/g), which is a nalidixic acid-resistant strain of ATCC 14028 used as a parental strain in this study, during 4 to 14 days after infection in a report by Rivera-Chaves et al. [52]. This finding likely provides at least one of the reasons why the wild-type did not elicit inflammation in the cecum of CBA mice in our present experiments. Even among the CBA mice infected with the *S.* Typhimurium wild-type strain at 13 days after infection, severe colitis in the cecum was not found by gross pathology (S9A Fig). We also obtained CBA mice from another breeder, Japan Charles River, but we did not observe severe inflammation in the cecum of mice infected with *S.* Typhimurium at 13 days after infection (S9B Fig). C57BL/6 and BALB/c mice are highly resistant to the intestinal inflammation induced by *S.* Typhimurium. Since severe colitis is elicited in mice pretreated with streptomycin and in germ-free mice infected with *S.* Typhimurium, the microbiota contribute to the resistance to *S.* Typhimurium colonization in these mice [20,54]. Our present findings suggest that the CBA mice used in this study may have somehow acquired colonization resistance to *S.* Typhimurium during housing before we obtained them.

In summary, our results demonstrated that the *Salmonella* effectors GogA and GtgA, as well as PipA, have metalloprotease activity, but the metalloprotease activity of PipA did not function in the HeLa cells infected with *S.* Typhimurium at an early stage of infection. It appeared that PipA was not expressed in the HeLa cells at an early stage of infection, because the expression of PipA was induced under T3SS-2-inducing conditions, but not under T3SS-1-inducing conditions. We also showed that PipA is translocated into RAW cells through T3SS-2, and that the enzymatic activity of PipA is responsible for systemic infection of *S.* Typhimurium. Further studies are needed to characterize the individual effectors identified in this study and thereby clarify the role played by the blocking of NF-κB signaling in *S.* Typhimurium infection.

## Supporting information

**S1 Fig. Confirmation of the expression of each effector protein in HeLa cells transfected with pEGFP-effector plasmid.** (A) Representative immunoblots of the EGFP-effector fusion protein in HeLa cells transfected with pEGFP-C1 or pEGFP-effector plasmids. HeLa cells were transfected with pEGFP-C1 or the indicated pEGFP-effector fusion plasmid. After 48 hrs of transfection, cells were lysed with $1 \times$ SDS sample buffer. Total proteins in the whole lysate were separated by SDS-PAGE, and EGFP or the EGFP fusion proteins were detected with immunoblotting using an anti-EGFP antibody. *Arrows*: The EGFP or EGFP-effector fusion proteins (α-GFP antibody). GAPDH was used as a loading control. (B) Fluorescent microscopy images of HeLa cells transfected with the indicated pEGFP-fusion plasmid (green), which was not detected by immunoblotting (effectors indicated in boldface in the panel A). HeLa cells were transfected with pEGFP-C1 or the indicated pEGFP-effector fusion plasmid. After 48 hrs of transfection, cells were visualized using a fluorescence microscope.
(PDF)

**S2 Fig. Confirmation of the expression of each effector-FLAG protein from the plasmid in the Δ*gogA*Δ*gtgA* strain.** The indicated bacterial strains were incubated for 15–20 hrs. Overnight bacterial cultures were diluted 1:33 in LB and incubated for 3 hrs in LB containing 0.3M NaCl and 1 μM Isopropyl β-D-1-thiogalactopyranoside (IPTG) to induce expressions of T3SS-1 or the effector gene from a *tac* promoter on the plasmids, respectively. The FLAG-fusion proteins were detected by immunoblotting using an anti-FLAG antibody.
(PDF)

**S3 Fig.** *S.* **Typhimurium cleaves NF-κB p65 in HeLa cells at an early, but not a late stage of infection.** HeLa cells were infected with the indicated *S.* Typhimurium strains and the genta-mycin killing assay was performed. After 2 or 16 hrs of infection, the cells were detached with TrypLE Select and the cytoplasmic fraction was isolated. The full-length p65 (Full p65) or cleaved p65 in the cytoplasmic fraction was detected by immunoblotting using an anti-p65 antibody. Immunoblots are representative of at least three independent experiments. (PDF)

**S4 Fig. GogA, GtgA, and PipA block the translocation of NF-κB p65 into the nucleus.** Fluo-rescent microscopy images of HeLa cells transfected with the indicated pEGFP-fusion plasmid are shown. HeLa cells were transfected with the indicated pEGFP fusion plasmid (green). After 48 hrs, the cells were stimulated with TNF-α (10 ng/ml) and then further cultured for 30 min and treated with anti-NF-κB subunit p65 antibodies (red) and DAPI (blue) to stain the nuclei. *White arrows*: The cells inhibiting the translocation of p65 into the nucleus by GogA, GtgA, or PipA. Scale bar, 10 μm. (PDF)

**S5 Fig. GogA, GtgA, or PipA cleave NF-κB p65.** Representative immunoblots of p65 in HEK293T cells transfected with the pEGFP-C1 or pEGFP effectors from three independent experiments. *Arrows*: Full p65 or cleaved p65 (α-p65 antibody), or EGFP or EGFP-effector fusion proteins (α-GFP antibody). Histone H3 and GAPDH were used as a loading control in the nucleus and the cytoplasm fraction, respectively. (PDF)

**S6 Fig. Genetic organization around the PipA family effector protein-encoded genes on the *S.* Typhimurium-specific prophages Gifsy-1 and Gifsy-2, and the pathogenicity island SPI-5.** The initial nucleotide on the *gogA*, *gtgA*, or *pipA* gene is indicated as +1. Arrows and the numbers above/beyond the arrows show the gene coding sequence (CDS) and the CDS names annotated for the genome of *S.* Typhimurium ATCC 14028, respectively. The % indi-cates the homology of the DNA sequence enclosed by dotted lines. (PDF)

**S7 Fig. GogA, GtgA, and PipA do not affect the virulence of *S.* Typhimurium in CBA mice.** CBA mice were infected intragastrically with 1×109 CFU of the indicated *S.* Typhimurium strains. (**A**) Bacterial numbers recovered from the colon contents, mesenteric lymph nodes (MLNs), and spleen at 4 days after infection. Individual data are shown as a scatter plot, and bars are the mean. (**B**) Histopathological changes were scored in sections of the cecum at 4 days after infection. Each bar represents the combined scoring results for a single mouse. (PDF)

**S8 Fig. Cleaved p65 was not detected in HeLa cells infected with a T3SS-1-deficient mutant (*invA*::pEP185.2, T1) or a Δ*gogA*Δ*gtgA* mutant of *S.* Typhimurium, or the two clinical iso-late stains, S72 and 1305–96, of *S.* Enteritidis.** The cleavage of p65 in HeLa cells infected with the indicated *S.* Typhimurium or *S.* Enteritidis strain was detected by immunoblotting using an anti-p65 antibody. (PDF)

**S9 Fig. Gloss pathology of the cecum of Nramp1+/+ mice infected with *Salmonella*.** (A) Cecum of CBA mice infected with the *S.* Typhimurium wild-type strain or the Δ*gogA*Δ*gtgA*Δ-*pipA* mutant at 13 days after infection. (B) Cecum of CBA mice from Charles River Japan infected with the *S.* Typhimurium wild-type strain or the Δ*gogA*Δ*gtgA*Δ*pipA* mutant at 4 days

after infection.
(PDF)

**S1 Table. Bacterial strains and plasmids used in this study.**
(PDF)

**S2 Table. Plasmids used in this study.**
(PDF)

**S3 Table. Nucleotide primers used in this study.**
(PDF)

**S1 Raw images.**
(PDF)

## Acknowledgments

We thank Drs. Toru Tobe and Hiroshi Ashida for providing plasmids. We also thank Shunta Ohtsuka, Kohei Harai, Manami Kimura, Sayaka Ohnaka, Miwa Matsumura, Asuka Tsukada, and Mayu Ohta for technical assistance.

## Author Contributions

**Conceptualization:** Takeshi Haneda, Tsuyoshi Miki.

**Funding acquisition:** Takeshi Haneda, Nobuhiko Okada.

**Investigation:** Momo Takemura, Takeshi Haneda, Hikari Idei.

**Resources:** Tsuyoshi Miki, Nobuhiko Okada.

**Supervision:** Nobuhiko Okada.

**Validation:** Momo Takemura, Takeshi Haneda, Hikari Idei.

**Visualization:** Momo Takemura, Takeshi Haneda, Hikari Idei.

**Writing – original draft:** Takeshi Haneda.

**Writing – review & editing:** Takeshi Haneda.

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
