## [Decision Letter · Decision Letter 0]

1 Feb 2021

PONE-D-20-38497

A *Salmonella* Type III effector, PipA, works in a different manner than the PipA family effectors GogA and GtgA

PLOS ONE

Dear Dr. Haneda,

First, let me apologize for the delay in getting your manuscript evaluated!  As you can see from their comments provided below, both reviewers expressed the opinion that the work described in the paper provides information that is important to the field. But they also point out numerous issues that need to addressed before the manuscript will be considered suitable for publication in *PLOS ONE*. Thus, I am going to ask that you submit a revised version of  the manuscript that adequately and appropriately addresses all of  the concerns raised by both of  these reviewers.

We look forward to receiving your revised manuscript!

Sincerely,

R. Martin Roop II, Ph.D.

Academic Editor

PLOS ONE

Journal Requirements:

2.PLOS ONE now requires that authors provide the original uncropped and unadjusted images underlying all blot or gel results reported in a submission’s figures or Supporting Information files. This policy and the journal’s other requirements for blot/gel reporting and figure preparation are described in detail at https://journals.plos.org/plosone/s/figures#loc-blot-and-gel-reporting-requirements and https://journals.plos.org/plosone/s/figures#loc-preparing-figures-from-image-files. When you submit your revised manuscript, please ensure that your figures adhere fully to these guidelines and provide the original underlying images for all blot or gel data reported in your submission. See the following link for instructions on providing the original image data: https://journals.plos.org/plosone/s/figures#loc-original-images-for-blots-and-gels.

Reviewers' comments:

Reviewer's Responses to Questions

**Comments to the Author**

1. Is the manuscript technically sound, and do the data support the conclusions?

Reviewer #1: Partly

Reviewer #2: Yes

2. Has the statistical analysis been performed appropriately and rigorously? 

Reviewer #1: Yes

Reviewer #2: Yes

3. Have the authors made all data underlying the findings in their manuscript fully available?

Reviewer #1: Yes

Reviewer #2: Yes

4. Is the manuscript presented in an intelligible fashion and written in standard English?

Reviewer #1: Yes

Reviewer #2: Yes

5. Review Comments to the Author

Reviewer #1: Please see enclosed the review of “A Salmonella Type III effector, PipA, works in a different manner than the PipA family effectors GogA and GtgA.” This well-written manuscript demonstrates that purified GogA, GtgA, and PipA reduce NF-kB signaling in tissue-cultured epithelial cells due to cleavage of NF-kB p65 subunit but only GogA and GtgA are required for the cleavage in these cells, likely due to secretion through the T3SS-1. While the PipA protein can cleave NF-kB p65, it is not active during infection with S. Typhimurium, likely due to secretion through the T3SS-2. The roles of PipA, GogA, and GtgA appear to be non-redundant during murine infection. The methodology lack details and/or clarity and there are numerous discrepancies noted below that require clarification. Since in vitro culture conditions can have a great impact on expression of T3SS-1 and T3SS-2 genes and resulting impact on host-pathogen interactions, a complete description of the methodology used for the reported work will facilitate understanding of the presented work as well as placement into context of prior work on the same genes and processes.

Major points:

For all figures: The description of each figure panel in the legend would improve the readers’ understanding of the figures as stand-alone data. Furthermore, statement of the number of technical and biological repeats each data point represents would allow for improved assessment of experimental rigor.

The authors state (Lines 263, 273, 323-325, 342, 493) that PipA does not dampen NF-kB production, but this statement is not supported by the data presented in Figures 2D and 3C, which demonstrate that PipA can compensate for loss-of-function of both GtgA and GogA. Please clarify.

The implication of PipA as a T3SS-2 effector comes from secretion assays (Figure 4C) as well as intracellular replication and cytotoxicity assays. Furthermore, prior work suggests that PipA is expressed in T3SS-2-inducing conditions (reference 45). However, the T1 PipA mutant behaves as the T1 mutant, not as a T1T2 mutant, suggesting no role for PipA in T3SS-2 function in RAW cells. Establishing the role of PipA in a ∆T3SS-2 mutant would more closely link PipA to the T3SS-2 function through genetic means. Furthermore, by linking PipA with T3SS-2, evaluation of PipA activity in HeLa cells at 4hpi is too early, as the T3SS-2 is not active at that time. Evaluation of PipA activity on NF-kB cleavage at a time when T3SS-2 is expressed would improve the clarity of the argument regarding PipA and T3SS-2 function. Furthermore, amendment of the discussion regarding the function of PipA in HeLa cells during early infection would clarify the discrepancies between overexpression and infection conditions in these cells.

Lines 309-313: The results presented in Figure 2C demonstrate a role for GtgA alone in NF-kB suppression, but GogA has no effect on its own, although there appears to be an additive effect of GogA and GtgA. These results merit discussion as to potential redundancy/lack thereof of these two effector proteins.

Minor points:

Lines 102: The authors mention a “slight modification…” of prior scoring system. Please describe the modification.

Line 138: pFLAG-CTC is not listed in S2 table.

Lines 202-203: Please describe culture conditions for stimulation of T3SS-1 and T3SS-2 with respect to osmolarity, oxygenation, and media composition. LPM acronym should be spelled out.

Lines 254-5: The details regarding TNF-a stimulation and duration conflict with those stated in the methods (lines 164-5). Please clarify.

Lines 268-9: The timing of NF-kB measurement relative to S. Typhimurium infection conflict with that in line 171. Furthermore, please describe the culture conditions for induction of T3SS-1 (line 170).

Lines 316-318 and Figure 2D: The methodology for induction of gene expression is not provided in the text. Please correct.

Figure 4A and 4B: Y-axis indicates normalization of target to gapdh, which is not a bacterial gene. Please correct.

Lines 378-381: Please indicate how induction of gogA/gtgA and pipA were determined. It is unclear from the text and figure legend what comparators were used to establish induction vs. no change in gene expression in a given condition.

Lines 412, 416, and others. Mice have one cecum, therefore it is correct to refer to mouse cecum, rather than mouse ceca. Please correct.

Line 421: the text mentions colonization in the cecum, but methods and figure axes indicate colonization in colon contents. Please clarify.

Figure 6: Please indicate whether the CI calculation indicates mutant/WT or vice versa to improve the readers’ understanding of the presented data.

supporting information: The legends for Figures S2-5 do not match the figures. Similarly, there are no figure legends for Figures S7 or S8. Please correct.

Reviewer #2: The manuscript by Takemura et al. identifies seven S. Typhimurium effectors that when overexpressed in HeLa cells, inhibit NF-kB signaling. Although effectors of the SseK and PipA family have previously been reported to target NF-kB activity, Takemura et al. here report for the first time that the effector SteE may also contribute to inhibiting NF-kB signaling. Using a series of Salmonella mutants and complemented strains the authors show that effector mediated inhibition of NF-kB by the PipA family of zinc metalloproteases GogA, GtgA and PipA cleaves the p69 subunit in a manner dependent on the induction of two Salmonella T3SS (T3SS-1 and T3SS-2 for GogA and GtgA and T3SS-2 for PipA).

The authors also sought to recapitulate a previously reported enhanced colitis induced by the gogA gtgA pipA S. Typhimurium mutant in an Slc11a1+/+ animal genetic background but were unsuccessful likely due to increased colonization resistance by the microbiota of their animals. Takemura et al., do however find that PipA protease activity does enhance systemic dissemination of S. Typhimurium in accordance with a previous observation by Knodler et al., (2002) that PipA is required for virulence in BALB/c mice.

The experiments are well designed and rigorously performed and statistical methods applied are in accordance with standards in the field. Clarity in the manuscript could be improved by addressing the following specific points:

1. Line 325: More specific language should be used to describe the cells referenced in this line. It is not clear if the authors mean the gogA gtgA Salmonella mutant cells or HeLa cells.

2. Line 396: Here the T1 PipA strain is first introduced and it’s genotype (invA::pEP185.1 pipA::Km) should be included (as it is in line 443) so as to not confuse it with a strain that overexpresses PipA.

3. Line 442: The complete strain name (C57BL/6) of the black 6 animal should be used instead of the term B6.

4. The rational for using the invA mutant background strains for the animal infections for Figure 6 should be briefly explained in the text.

5. Line 479: The authors reference the spiB mutant yet Figure S7 indicates that the ssaV mutant was used. Please revise the text or figure to accurately reflect which strain was used in this experiment.

6. The figure legends for S7 and S8 are missing from the text. Please add figure legends for these supplemental figures.

6. PLOS authors have the option to publish the peer review history of their article (what does this mean?). If published, this will include your full peer review and any attached files.

Reviewer #1: No

Reviewer #2: No

---

## [Author Response · Author response to Decision Letter 0]

16 Feb 2021

Reviewer #1: Please see enclosed the review of “A Salmonella Type III effector, PipA, works in a different manner than the PipA family effectors GogA and GtgA.” This well-written manuscript demonstrates that purified GogA, GtgA, and PipA reduce NF-kB signaling in tissue-cultured epithelial cells due to cleavage of NF-kB p65 subunit but only GogA and GtgA are required for the cleavage in these cells, likely due to secretion through the T3SS-1. While the PipA protein can cleave NF-kB p65, it is not active during infection with S. Typhimurium, likely due to secretion through the T3SS-2. The roles of PipA, GogA, and GtgA appear to be non-redundant during murine infection. The methodology lack details and/or clarity and there are numerous discrepancies noted below that require clarification. Since in vitro culture conditions can have a great impact on expression of T3SS-1 and T3SS-2 genes and resulting impact on host-pathogen interactions, a complete description of the methodology used for the reported work will facilitate understanding of the presented work as well as placement into context of prior work on the same genes and processes.

Major points:

For all figures: The description of each figure panel in the legend would improve the readers’ understanding of the figures as stand-alone data. Furthermore, statement of the number of technical and biological repeats each data point represents would allow for improved assessment of experimental rigor.

RESPONSE: We appreciate the reviewer’s comment. We revised all figure legends for clarity, with the exception of Fig 6.

The authors state (Lines 263, 273, 323-325, 342, 493) that PipA does not dampen NF-kB production, but this statement is not supported by the data presented in Figures 2D and 3C, which demonstrate that PipA can compensate for loss-of-function of both GtgA and GogA. Please clarify.

RESPONSE: We thank the reviewer for pointing this out. As shown in Fig 4A, PipA is not expressed in the early stage of infection of S. Typhimurium. This is the reason why the ∆gogA∆gtgA strain does not suppress the NF-κB activation and cleave p65 in HeLa cells infected with S. Typhimurium, although this strain has a pipA gene (Figs 2C and 3C). However, the ∆gogA∆gtgA (pPipA) strain, which overexpresses PipA from the plasmid, dampens the NF-κB activation. These facts suggest that the PipA protein would potentially dampen the NF-κB activation by cleaving p65. We now address this in the Discussion section (lines 528–531).

The implication of PipA as a T3SS-2 effector comes from secretion assays (Figure 4C) as well as intracellular replication and cytotoxicity assays. Furthermore, prior work suggests that PipA is expressed in T3SS-2-inducing conditions (reference 45). However, the T1 PipA mutant behaves as the T1 mutant, not as a T1T2 mutant, suggesting no role for PipA in T3SS-2 function in RAW cells. Establishing the role of PipA in a ∆T3SS-2 mutant would more closely link PipA to the T3SS-2 function through genetic means. Furthermore, by linking PipA with T3SS-2, evaluation of PipA activity in HeLa cells at 4hpi is too early, as the T3SS-2 is not active at that time. Evaluation of PipA activity on NF-kB cleavage at a time when T3SS-2 is expressed would improve the clarity of the argument regarding PipA and T3SS-2 function. Furthermore, amendment of the discussion regarding the function of PipA in HeLa cells during early infection would clarify the discrepancies between overexpression and infection conditions in these cells.

RESPONSE: We appreciate the reviewer’s comments. We added data on the NF-κB activity of HeLa cells infected with the wild-type or the ∆gogA∆gtgA∆pipA∆sseK1∆sseK2∆sseK3∆steE strain after 20 hrs of infection (Fig 2A). These data show that effectors other than the five identified effectors might attenuate the NF-κB signaling pathway at 20 hrs after infection. Previous studies have shown that other T3SS-2 effectors, i.e., AvrA, GogB, SpvD, SseL, SspH1 or SteA, inhibit the NF-κB activation (references 8–12, 47). These effectors inhibit the NF-κB signaling pathway in a way that is different from cleaving p65. Interestingly, in addition to PipA, most of these other effectors also have no effect on intracellular survival and induction of cell death in RAW cells infected with S. Typhimurium. These data might indicate that these T3SS-2 effectors have overlapping functions in the S. Typhimurium-infected cells. A mutant in which all of these effectors are deleted will be needed to understand the role of the inhibition of the NF-κB signaling in the late stage of Salmonella infection. We revised the Discussion section to incorporate these points (lines 515¬¬–525).

We also added the immunoblotting data related to the detection of cleaved p65 in HeLa cells after 16 hrs of infection (Fig S3). However, we did not observe cleaved p65 in HeLa cells infected with the wild-type or any of the mutants used in this experiment. This data may support the hypothesis that PipA digests another target in the late stage of Salmonella infection (lines 561–562). 

Lines 309-313: The results presented in Figure 2C demonstrate a role for GtgA alone in NF-kB suppression, but GogA has no effect on its own, although there appears to be an additive effect of GogA and GtgA. These results merit discussion as to potential redundancy/lack thereof of these two effector proteins.

RESPONSE: We appreciate the reviewer’s comment. The level of NF-κB activity of HeLa cells infected with the ∆gogA strain is significantly higher than that of the wild type strain, but that of HeLa cells infected with the ∆gtgA strain is not. However, the NF-κB activity of the cells infected with the ∆gtgA or ∆gogA strain is lower than that of cells infected with the ∆gogA∆gtgA or ∆gogA∆gtgA∆pipA strain, both of which are incapable of p65 cleavage. We also showed that not only the ∆gtgA and ∆pipA strains but also the ∆gogA strain cleaves NF-κB p65 (Fig 3A). This indicates that the level of the NF-κB activity of the cells infected with the ∆gtgA strain is sufficient to show cleavage of p65. We now address this in the Discussion section (lines 502–507).

Minor points:

Lines 102: The authors mention a “slight modification…” of prior scoring system. Please describe the modification.

RESPONSE: We added a description of our modification in the Materials and Methods section as requested (lines 103–107).

Line 138: pFLAG-CTC is not listed in S2 table.

RESPONSE: We added pFLAG-CTC and its derivatives in Table S2.

Lines 202-203: Please describe culture conditions for stimulation of T3SS-1 and T3SS-2 with respect to osmolarity, oxygenation, and media composition. LPM acronym should be spelled out.

RESPONSE: We added the culture conditions in detail in the Materials and Methods section as requested (lines 208–210).

Lines 254-5: The details regarding TNF-a stimulation and duration conflict with those stated in the methods (lines 164-5). Please clarify.

RESPONSE: We corrected the duration of treatment with TNF-α in the Materials and Methods section (line 170).

Lines 268-9: The timing of NF-kB measurement relative to S. Typhimurium infection conflict with that in line 171. Furthermore, please describe the culture conditions for induction of T3SS-1 (line 170).

RESPONSE: We revised the timing of measurement of the NF-κB activity (line 176). We also added the culture conditions for induction of T3SS-1 (line 175) .

Lines 316-318 and Figure 2D: The methodology for induction of gene expression is not provided in the text. Please correct.

RESPONSE: We added the methodology for induction of gene expression in the Materials and Methods section and the legend of Fig 2D (lines 208–210 and line 284).

Figure 4A and 4B: Y-axis indicates normalization of target to gapdh, which is not a bacterial gene. Please correct.

RESPONSE: We changed gapdh to gyrA on the Y-axis in Figs 4A and 4B.

Lines 378-381: Please indicate how induction of gogA/gtgA and pipA were determined. It is unclear from the text and figure legend what comparators were used to establish induction vs. no change in gene expression in a given condition.

RESPONSE: We indicated how induction of these genes was determined (lines 393–395).

Lines 412, 416, and others. Mice have one cecum, therefore it is correct to refer to mouse cecum, rather than mouse ceca. Please correct.

RESPONSE: We corrected mouse ceca to mouse cecum.

Line 421: the text mentions colonization in the cecum, but methods and figure axes indicate colonization in colon contents. Please clarify.

RESPONSE: We corrected cecum to colon contents (line 442).

Figure 6: Please indicate whether the CI calculation indicates mutant/WT or vice versa to improve the readers’ understanding of the presented data.

RESPONSE: We changed each graph title in Fig 6 from WT vs. mutant to mutant vs. WT. And we also added mutant/WT to the Y-axis.

supporting information: The legends for Figures S2-5 do not match the figures. Similarly, there are no figure legends for Figures S7 or S8. Please correct.

RESPONSE: We revised the legends for Figs S2–5 (Fig S2 and S4–6 in the revised supporting information) and added the legends for Fig S7 and S8 (Figs S8 and S9 in the revised supporting information). 

 

Reviewer #2: The manuscript by Takemura et al. identifies seven S. Typhimurium effectors that when overexpressed in HeLa cells, inhibit NF-kB signaling. Although effectors of the SseK and PipA family have previously been reported to target NF-kB activity, Takemura et al. here report for the first time that the effector SteE may also contribute to inhibiting NF-kB signaling. Using a series of Salmonella mutants and complemented strains the authors show that effector mediated inhibition of NF-kB by the PipA family of zinc metalloproteases GogA, GtgA and PipA cleaves the p69 subunit in a manner dependent on the induction of two Salmonella T3SS (T3SS-1 and T3SS-2 for GogA and GtgA and T3SS-2 for PipA).

The authors also sought to recapitulate a previously reported enhanced colitis induced by the gogA gtgA pipA S. Typhimurium mutant in an Slc11a1+/+ animal genetic background but were unsuccessful likely due to increased colonization resistance by the microbiota of their animals. Takemura et al., do however find that PipA protease activity does enhance systemic dissemination of S. Typhimurium in accordance with a previous observation by Knodler et al., (2002) that PipA is required for virulence in BALB/c mice.

The experiments are well designed and rigorously performed and statistical methods applied are in accordance with standards in the field. Clarity in the manuscript could be improved by addressing the following specific points:

1. Line 325: More specific language should be used to describe the cells referenced in this line. It is not clear if the authors mean the gogA gtgA Salmonella mutant cells or HeLa cells.

RESPONCSE: We appreciate the reviewer’s comment. We changed the cells to the ∆gogA∆gtgA strain to clarify our meaning in this sentence (line 356).

2. Line 396: Here the T1 PipA strain is first introduced and it’s genotype (invA::pEP185.1 pipA::Km) should be included (as it is in line 443) so as to not confuse it with a strain that overexpresses PipA.

RESPONSE: We changed T1 PipA to invA::pEP185.1 pipA::Km (line 416).

3. Line 442: The complete strain name (C57BL/6) of the black 6 animal should be used instead of the term B6.

RESPONSE: We changed B6 to C57BL/6 (line 466).

4. The rational for using the invA mutant background strains for the animal infections for Figure 6 should be briefly explained in the text.

RESPONSE: We added the reason why the strains were used in the mouse experiment (lines 462–465).

5. Line 479: The authors reference the spiB mutant yet Figure S7 indicates that the ssaV mutant was used. Please revise the text or figure to accurately reflect which strain was used in this experiment.

RESPONSE: We revised the text to indicate that the ssaV::Cm mutant was used (line 509).

6. The figure legends for S7 and S8 are missing from the text. Please add figure legends for these supplemental figures.

RESPONSE: We added figure legends for Figs. S7 and S8 (Figs S8 and S9 in the revised supporting information).

---

## [Decision Letter · Decision Letter 1]

5 Mar 2021

PONE-D-20-38497R1

A *Salmonella* Type III effector, PipA, works in a different manner than the PipA family effectors GogA and GtgA

PLOS ONE

Dear Dr. Haneda,

Thank you for your thorough attention to the concerns raised by the reviewers!  But as you can see from Reviewer 1's comments there are still a few minor points that need to be addressed/clarified. Once these have been dealt with, your paper will be considered suitable for publication in *PLOS ONE*.

Please submit your revised manuscript by June 4, 2021. But to be honest, I don't think that addressing these points will take that much time. But if  you will need more time than this to complete your revisions, please reply to this message or contact the journal office at plosone@plos.org. Please include the following items when submitting your revised manuscript:

I look forward to seeing your revised manuscript. I also appreciate your patience with the review process!

Sincerely,,

R. Martin Roop II, Ph.D.

Academic Editor

PLOS ONE

Journal Requirements:

Reviewers' comments:

Reviewer's Responses to Questions

**Comments to the Author**

1. If the authors have adequately addressed your comments raised in a previous round of review and you feel that this manuscript is now acceptable for publication, you may indicate that here to bypass the “Comments to the Author” section, enter your conflict of interest statement in the “Confidential to Editor” section, and submit your "Accept" recommendation.

Reviewer #1: (No Response)

Reviewer #2: All comments have been addressed

2. Is the manuscript technically sound, and do the data support the conclusions?

Reviewer #1: Yes

Reviewer #2: Yes

3. Has the statistical analysis been performed appropriately and rigorously? 

Reviewer #1: Yes

Reviewer #2: Yes

4. Have the authors made all data underlying the findings in their manuscript fully available?

Reviewer #1: Yes

Reviewer #2: Yes

5. Is the manuscript presented in an intelligible fashion and written in standard English?

Reviewer #1: Yes

Reviewer #2: Yes

6. Review Comments to the Author

Reviewer #1: The revised manuscript entitled “A Salmonella type III effector, PipA, works in a different manner than the PipA family effectors GogA and GtgA” has addressed most reviewer comments and is improved in clarity. The added data support the authors’ hypothesis that PipA, while capable of cleavage of p65, is unlikely to perform this function in HeLa cells during early infection. However, there are a few points that require clarification, as described below:

Minor points:

1. Plasmids used for expression of EGFP-fusion proteins in Figures 1 and S1 are incompletely described. Tables S2 and S3 describe plasmids containing GtgA, GogA, and PipA only. Please indicate primers/plasmids used to generate data for Figures 1 and S1, or reference study producing the constructs.

2. Lines 169-70 and 261-2 and 32 in supporting information (Fig S4 legend) conflict as to TNF-a stimulation.

3. Data regarding bacterial colonization in the cecum are not presented in Figure 5A (line 444). Please correct.

4. Lines 508-511 provide new experimental results in support of data in Fig 4. I recommend moving these data to results section in support of T3SS-1 secretion as important for early p65 cleavage.

5. Lines 526-531. There is an apparent discrepancy between sentences in 526-7 and 530-1. I suggest clarification in lines 530-1 that the PipA protein can cleave p65 but does not appear to do so at the times measured in HeLa cells during S. Typhimurium infection.

Reviewer #2: The manuscript by Takemura et al. identifies seven S. Typhimurium effectors that when overexpressed in HeLa cells, inhibit NF-kB signaling.

The experiments are well designed and rigorously performed and statistical methods applied are in accordance with standards in the field.

Authors have also adequately addressed all comments.

7. PLOS authors have the option to publish the peer review history of their article (what does this mean?). If published, this will include your full peer review and any attached files.

Reviewer #1: No

Reviewer #2: No

---

## [Author Response · Author response to Decision Letter 1]

6 Mar 2021

Reviewer #1: The revised manuscript entitled “A Salmonella type III effector, PipA, works in a different manner than the PipA family effectors GogA and GtgA” has addressed most reviewer comments and is improved in clarity. The added data support the authors’ hypothesis that PipA, while capable of cleavage of p65, is unlikely to perform this function in HeLa cells during early infection. However, there are a few points that require clarification, as described below:

Minor points:

1. Plasmids used for expression of EGFP-fusion proteins in Figures 1 and S1 are incompletely described. Tables S2 and S3 describe plasmids containing GtgA, GogA, and PipA only. Please indicate primers/plasmids used to generate data for Figures 1 and S1, or reference study producing the constructs.

Response: We added primers and plasmids as requested.

2. Lines 169-70 and 261-2 and 32 in supporting information (Fig S4 legend) conflict as to TNF-a stimulation.

Response: We corrected “20 ng/ml” to “10 ng/ml” in line 169.

3. Data regarding bacterial colonization in the cecum are not presented in Figure 5A (line 444). Please correct.

Response: We corrected “cecum” to “colon contents” in line 444.

4. Lines 508-511 provide new experimental results in support of data in Fig 4. I recommend moving these data to results section in support of T3SS-1 secretion as important for early p65 cleavage.

Response: We appreciate the reviewer’s comment. However, S8 Fig. should be left as is if possible. In Fig 4, it shows that PipA is a T3SS-2 effector rather than that GogA and GtgA are T3SS-1 effectors. On the other hand, S8 Fig shows not only that GogA and GtgA cleave p65 in T3SS-1 dependent manner, but also that the inactivation of NF-κB by the GogA and GtgA are the S. Typhimurium-specific phenotype in Salmonella. It has been published that GogA and GtgA function as T3SS-1 effectors and the genes encoding GogA and GtgA are located on prophages specifically integrated in the genome of S. Typhimurium. Thus, the data on S8 Fig. support Figs 2 and 3, and the previous reports. 

5. Lines 526-531. There is an apparent discrepancy between sentences in 526-7 and 530-1. I suggest clarification in lines 530-1 that the PipA protein can cleave p65 but does not appear to do so at the times measured in HeLa cells during S. Typhimurium infection.

Response: We revised the sentence as suggested.

---

## [Editor Report · Decision Letter 2]

9 Mar 2021

A *Salmonella* Type III effector, PipA, works in a different manner than the PipA family effectors GogA and GtgA

PONE-D-20-38497R2

Dear Dr. Haneda,

Thanks for the rapid turnaround of  the manuscript! I'm pleased to inform you that your paper has now been judged scientifically suitable for publication and will be formally accepted for publication once it meets all outstanding technical requirements.

Sincerely,

R. Martin Roop II, Ph.D.

Academic Editor

PLOS ONE
---

## [Editor Report · Acceptance letter]

11 Mar 2021

PONE-D-20-38497R2 

A *Salmonella* Type III effector, PipA, works in a different manner than the PipA family effectors GogA and GtgA 

Dear Dr. Haneda:

I'm pleased to inform you that your manuscript has been deemed suitable for publication in PLOS ONE. Congratulations! Your manuscript is now with our production department. 

Kind regards, 

on behalf of

Dr. Roy Martin Roop II 

Academic Editor

PLOS ONE